# Trends in Long COVID Symptoms in Japanese Teenage Patients

**DOI:** 10.3390/medicina59020261

**Published:** 2023-01-29

**Authors:** Yasue Sakurada, Yuki Otsuka, Kazuki Tokumasu, Naruhiko Sunada, Hiroyuki Honda, Yasuhiro Nakano, Yui Matsuda, Toru Hasegawa, Kanako Ochi, Hideharu Hagiya, Keigo Ueda, Hitomi Kataoka, Fumio Otsuka

**Affiliations:** 1Department of General Medicine, Okayama University Graduate School of Medicine, Dentistry and Pharmaceutical Sciences, Okayama 700-8558, Japan; 2Center for Education in Medicine and Health Sciences, Okayama University Graduate School of Medicine, Dentistry and Pharmaceutical Sciences, Okayama 700-8558, Japan

**Keywords:** Delta variant, schoolchildren, teenagers, Omicron variant, post-COVID-19 condition

## Abstract

*Background*: Since the start of the global pandemic of coronavirus disease 2019 (COVID-19), not only adults but also many children have suffered from it. However, the clinical characteristics of long COVID in children remain unclear. *Methods*: In this retrospective observational study conducted in a single facility, we reviewed the medical records of all long COVID patients who visited Okayama University Hospital from February 2021 to October 2022, and we compared the clinical characteristics of long COVID in teenagers (11 to 18 years of age) with those in adults. *Results*: Data for 452 long COVID patients including 54 teenagers (11.9%) were analyzed. Fatigue was the most frequent symptom in teenagers (55.6% of the patients) and also in adults. On the other hand, the percentage of teenagers who complained of headache, which was the second most frequent complaint, was significantly higher than the percentage of adults (35.2% vs. 21.9%, *p* < 0.05). A comparison of the frequencies of symptoms depending on the viral variant showed that fatigue and headache were predominant symptoms in the Omicron variant phase. Of the 50 teenagers who were enrolled in schools, 28 (56.0%) could not attend school due to long COVID symptoms. The most common symptoms as reasons for absence from school were fatigue (85.7% of the patients), headache (42.9%), and insomnia (32.1%). *Conclusions*: Attention should be paid to the symptoms of fatigue and headache in teenagers with long COVID.

## 1. Introduction

The global pandemic of coronavirus disease 2019 (COVID-19) started in 2019 and many mutant variants of the virus have since emerged [1,2]. Even after surviving the acute phase of COVID-19, approximately one-third of patients suffer prolonged symptoms [3,4]. Although these symptoms have been studied by many institutions including the World Health Organization [5], the mechanisms underlying the prolonged symptoms have remained unclear. Moreover, studies on long COVID in adolescents and children have been limited compared to studies in adults [6]. Advanced age has been reported to be a risk factor for long COVID in adults [7], but the association of age with long COVID in younger populations is not clear.

However, there have been increases in both the incidence of COVID-19 and prevalence of long COVID in children [8,9]. Improvement in the medical care system for long COVID in adults is of course an important issue for governments, but attention should also be paid to long COVID in children, who need to be cared for as they grow up from infants to toddlers to schoolchildren and to adolescents.

We established a COVID-19 aftercare outpatient clinic (CAC) in February of 2021, and we have been providing treatment for long COVID in children as well as adults for almost two years as a hub center in western Japan [9,10]. In this study, we compared the clinical characteristics of long COVID in teenagers with those in adults and revealed the trends caused by viral variant changes and the key issues related to school absence.

## 2. Patients and Methods

### 2.1. Study Design and Patients’ Characteristics

This study was an observational study that was performed retrospectively at a single facility. All long COVID patients who visited the CAC at Okayama University Hospital between 15 February 2021 and 31 October 2022 had their medical records evaluated. Long COVID was defined as symptoms that persist for more than four weeks after the onset of COVID-19, and all of the patients with long COVID were enrolled in this study. We obtained information on age, sex, severity of COVID-19, number of days between onset of COVID-19 and first CAC visit, number of COVID-19 vaccinations, clinical symptoms of post-COVID-19 condition (PCC), and school enrollment and attendance. We defined teenage patients as patients aged from 11 to 18 years and adult patients as patients who were 19 years of age or older. Patients who were enrolled in a school but were not able to attend the school were defined as patients absent from school. The severity of COVID-19 in the acute phase was categorized in accordance with the criteria established by the Japanese Ministry of Health, Labor, and Welfare [11]. Each patient underwent a complete face-to-face medical interview and examination by a physician in order to identify clinical symptoms of PCC.

### 2.2. Definitions of the Delta Variant and Omicron Variant Periods

According to our epidemiological classification of COVID-19 in Okayama Prefecture in Japan, [12] we divided the patients’ onset of COVID-19 into three groups: the Preceding phase is the period from the ancestral variant to the Alpha variant, before 18 July 2021; the Delta variant phase is the period from 19 July 2021 to 31 December 2021, when the Delta variant was dominant after the Alpha variant phase; and the Omicron variant phase is the period after 1 January 2022, when the Omicron variant was dominant in Okayama Prefecture [13].

### 2.3. Statistical Analysis

Stata SE, version 17, statistical software (StataCorp., College Station, TX, USA) was used for all analyses. The data were presented as number (%) for categorical variables and mean (standard deviation: SD) for continuous variables. After Skewness–Kurtosis tests for normality, we used Mann–Whitney U tests, *t* tests, and chi-squared tests for associations between non-normally distributed variables, between normally distributed variables, and between categorical variables, respectively. The threshold for significance was defined as * *p* < 0.05 and ** *p* < 0.01.

### 2.4. Ethical Approval

Patients who wished to opt out were given the opportunity to do so after information about the study was posted on our hospital’s wall and website. Since the data were anonymized, patients’ informed consent was not required. This study followed the Declaration of Helsinki and received approval from the Ethics Committee of Okayama University Hospital (No. 2105-030).

## 3. Results

Data for all of the 460 long COVID patients who visited our CAC during the study period were obtained from medical records. We excluded eight patients including four patients who were under ten years of age, one patient who had insufficient data, and three patients who were asymptomatic. The remaining 452 patients were firstly compared by dividing the patients into teenagers (54 patients, 11.9%) and adults (398 patients, 88.1%) (Figure 1). The mean ages of the teenagers (15.3 years) and the adults (42.9 years) were significantly different. The proportions of males and females, symptom onset phases, and periods from onset until the first visit were not significantly different between the two groups; however, the proportion of patients with moderate-to-severe illness during the acute phase was significantly lower in teenagers (0% vs. 19.4%, ** *p* < 0.01) (Table 1). The most frequent chief complaint in both groups was fatigue (55.6% vs. 61.1%, *p* = 0.438). In teenagers, the second most frequent chief complaint was headache, and the percentage of teenagers who complained of headache was significantly higher than the percentage of adults who complained of headache (35.2% vs. 21.9%, * *p* = 0.03). The percentages of patients who complained of other symptoms were not significantly different between the teenager and adult groups (Figure 2).

Changes in the major symptoms in teenage patients with long COVID in the three phases are shown in Figure 3. Dysosmia and dysgeusia, which were common symptoms in the Delta variant phase, almost disappeared in the Omicron variant phase. Instead, fatigue and headache became predominant symptoms in the Omicron variant phase.

Of the 54 teenage patients, 50 were enrolled in schools including elementary schools (4 patients, 8%), junior high schools (16 patients, 32%), and high schools (29 patients, 58%). Among the patients enrolled in schools, 28 patients (56%) were absent from school (Figure 1). Mean age, proportions of males and females, symptom onset phase, number of COVID-19 vaccinations, and number of patients enrolled in schools in the patients who were absent from school were not significantly different from those in the patients who were not absent from school (22 patients, 44%) (Table 2). The most common symptoms in teenage patients who were absent from school were fatigue (85.7% of the patients), headache (42.9%), and insomnia (32.14%) (Figure 4).

## 4. Discussion

In this study, we identified clinical characteristics of long COVID in teenagers. While previous studies showed that the prevalence of long COVID in children varied widely from a few percent to 70 percent [14], 12% of the long COVID patients in our study were teenagers. Although studies on the clinical characteristics of long COVID in children have been limited, fatigue was consistently the most frequent symptom in previous studies [15,16,17], which was consistent with the results of our study. As for headache, which was the second most frequent symptom in our study, there have been reports that headache does not often persist into the chronic phase and that its prevalence is varied [17]. In our study, we compared the frequencies of symptoms in teenagers with those in adults and found that headache, insomnia, and low-grade fever were the characteristic symptoms of long COVID in teenagers.

The symptoms of fatigue and headache were revealed to have become predominant in the Omicron variant phase. We found in a recent study that the number of adult patients complaining of fatigue was increasing during the Omicron variant phase, even in patients who did not have severe disease in the acute phase [13], a trend that was also found in children. On the other hand, we found that the frequencies of dysosmia and dysgeusia were decreasing, which was consistent with the results of a nationwide retrospective cohort study in Japan [18]. Although it has been reported that the Omicron variant produces less sequelae [19], the differences in long COVID symptoms depending on the variant of the virus are not well known, especially in children. Considering that headache and fatigue are directly related to a deterioration in the quality of life [20,21,22] and considering the possibility of long COVID developing to myalgic encephalomyelitis/chronic fatigue syndrome (ME/CFS) [23], the number of children who have problems in their daily life may have been increasing in the Omicron variant phase, and that would explain why there were many children unable to attend schools.

Indeed, it was shown in a study that children with COVID-19 had a significantly higher rate of school absenteeism than that in controls [24], and school absenteeism is becoming a social problem in the COVID-19 pandemic [25,26]. Various diseases might be causes of school absenteeism, whereas school absenteeism itself may further lead to other healthcare problems [27]. Although it was known even before the COVID-19 pandemic that some children suffered from such severe fatigue that they could not attend school [28], the number of children with fatigue may have been increasing during the COVID pandemic. Long COVID symptoms such as fatigue and headache may become reasons for increasing school absenteeism among schoolchildren. Although we previously revealed that fatigue including ME/CFS in adults with long COVID may be due to endocrine dysfunction represented by thyroid and adrenal hormonal effects and primary or secondary hypogonadism [29,30,31,32], the pathophysiological mechanisms of the various symptoms in children require further study. The effect of the vaccine on long COVID has been discussed [33], although the number of vaccinations was not significantly related to school absenteeism in this study.

Compared to various preceding studies based on online questionnaires [14], the identification of long COVID symptoms in this study may be more accurate because the symptoms were identified by face-to-face interviews. This study is unique in that we were able to perform a direct comparison of symptoms in children with those in adults because both the adults and children were seen by the same general physicians using the same evaluation tools as those proposed from a study conducted in Britain [34]. However, our study also has some limitations. First, the number of cases was small. Second, there could potentially be another patient population being seen by primary care physicians since only referred patients were included in this study. Third, the infected variants were defined by the time of infection, not by laboratory testing, and they may therefore not be strictly accurate.

## 5. Conclusions

In conclusion, we identified the clinical characteristics of long COVID in teenagers compared with those in adults. The symptoms of long COVID in teenagers differed depending on the viral variant, and the symptoms in teenagers were more prevalent than those in adults. Long COVID has a great impact on the social life of teenagers, making it difficult for them to attend school or engage in schoolwork, which may have serious and long-term effects. Thus, physicians need to comprehensively confront nonspecific symptoms such as symptoms of long COVID. Not only medical care in hospitals but also the utilization of social resources and collaboration with the community or schools are necessary for physicians to treat long COVID in children.

## Figures and Tables

**Figure 1 medicina-59-00261-f001:**
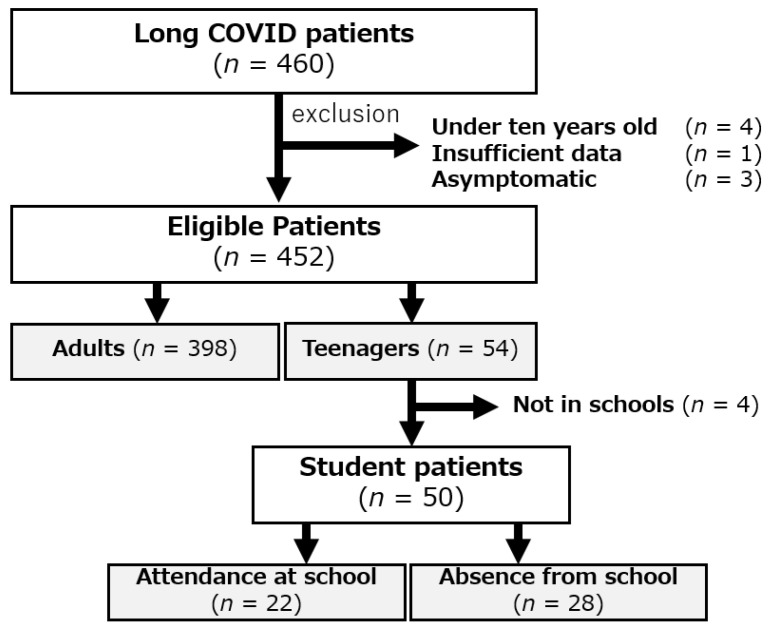
Patients’ characteristics. Among 460 long COVID patients, 452 were eligible and compared by dividing the patients into 54 teenagers and 398 adults. Fifty students enrolled in schools were investigated.

**Figure 2 medicina-59-00261-f002:**
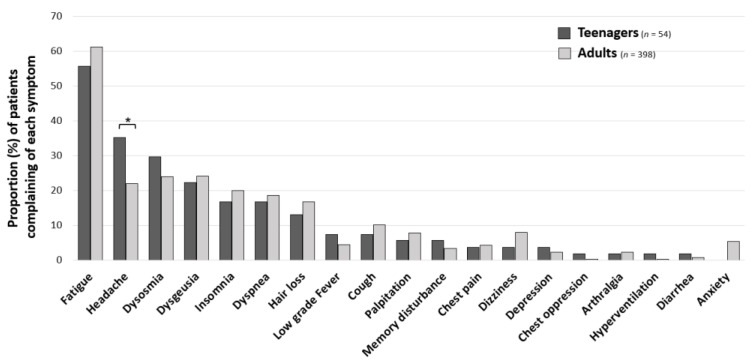
Differences between major symptoms in teenage patients and adult patients with long COVID. The percentages of patients who complained of each symptom are shown for teenagers (11 to 18 years old) and adults (19 years of age or more). Chi-squared tests were performed for the differences in percentages between the two groups. * *p* < 0.05 was considered statically significant.

**Figure 3 medicina-59-00261-f003:**
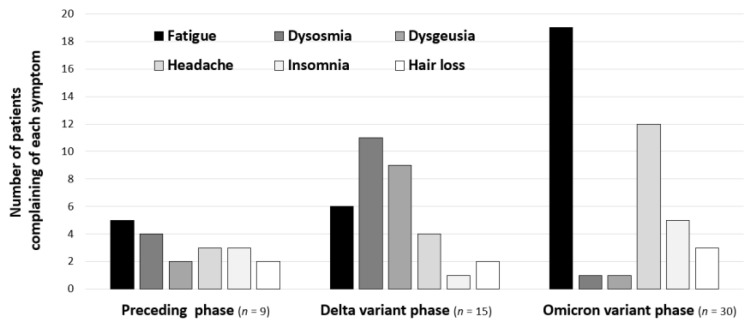
Variant-dependent changes in major symptoms in teenage patients with long COVID. The numbers of long COVID patients who complained of major symptoms are shown for the top six symptoms caused by infection with the variants, which were presumed on the basis of epidemiological aspects of COVID-19 in Okayama Prefecture in Japan: Preceding phase, before 18 July 2021; Delta variant phase, from 19 July 2021 to 31 December 2021; Omicron variant phase, after 1 January 2022.

**Figure 4 medicina-59-00261-f004:**
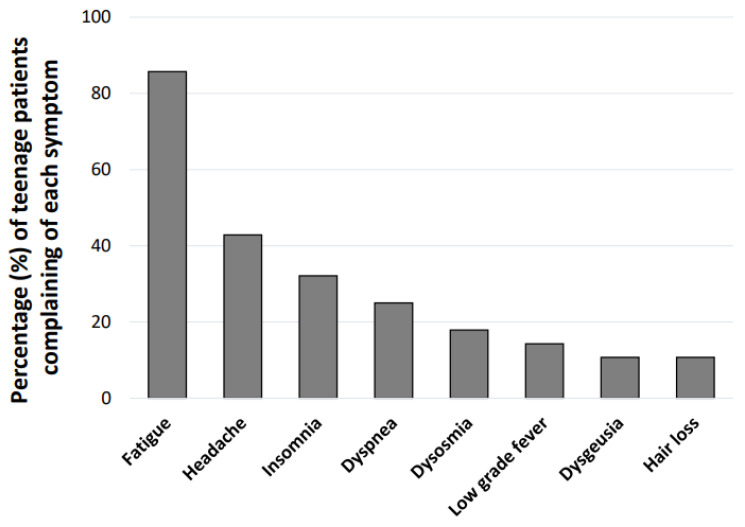
Reasons for absence from school in the teenage patients with long COVID. The percentage (%) of patients who complained of each major symptom is shown for the 28 teenage patients who were absent from school.

**Table 1 medicina-59-00261-t001:** Backgrounds of long COVID patients by their age categories.

	Teenagers (*n* = 54)	Adults (*n* = 398)	*p*-Value
**Age years, mean (SD)**			
	15.3 (1.8)	42.9 (13.7)	<0.001 ^(a)^ **
**Gender, *n* (%)**			
**Male**	23 (42.6)	183 (46)	0.639 ^(b)^
**Female**	31 (57.4)	215 (54.1)	
**Symptom onset phase, *n* (%)**			
**Preceding phase**	9 (16.7)	102 (25.6)	0.253 ^(b)^
**Delta variant phase**	15 (27.8)	117 (29.4)	
**Omicron variant phase**	30 (55.6)	179 (45)	
**Period from the onset until the first visit, *n* (%)**			
**<60 days**	14 (25.9)	129 (32.4)	0.289 ^(b)^
**60~90 days**	19 (35.2)	101 (25.4)	
**>90**	21 (38.9)	168 (42.2)	
**Severity of COVID-19 in acute phase, *n* (%)**			
**Mild**	54 (100)	321 (80.7)	<0.001 ^(b)^ **
**Moderate–Severe**	0 (0)	77 (19.4)	

The data were analyzed by using (a) the Mann–Whitney U test or (b) the chi-squared test; ** *p* < 0.01 was regarded as statistically significant.

**Table 2 medicina-59-00261-t002:** Backgrounds of student patients with long COVID.

	Attendance at School (*n* = 22)	Absence from School (*n* = 28)	*p*-Value
**Age years, mean (SD)**			
	14.7 (2)	15.4 (1.4)	0.556 ^(a)^
**Gender, *n* (%)**			
**Male**	11 (50)	10 (35.7)	0.310 ^(b)^
**Female**	11 (50)	18 (64.3)	
**Symptom onset phase, *n* (%)**			
**Preceding phase**	2 (9.1)	6 (21.4)	0.214 ^(b)^
**Delta variant phase**	9 (40.9)	6 (21.4)	
**Omicron variant phase**	11 (50)	16 (57.1)	
**Period from the onset until the first visit, *n* (%)**			
**<60 days**	4 (18.2)	10 (35.7)	0.159 ^(b)^
**<60 days**	4 (18.2)	10 (35.7)	0.159 ^(b)^
**60~90 days**	7 (31.8)	11 (39.3)	
**>90**	11 (50)	7 (25)	
**Vaccination, *n* (%)**			
**<2 times**	11 (50)	12 (42.9)	0.696 ^(b)^
**2 times**	10 (45.5)	13 (46.4)	
**>3 times**	1 (4.6)	3 (10.7)	
**School, *n* (%)**			
**Elementary school**	3 (13.6)	1 (3.6)	0.294 ^(b)^
**Junior high school**	8 (36.4)	8 (28.6)	
**High school**	11 (50)	19 (67.9)	

The data were analyzed by using (a) *t* tests or (b) chi-squared tests; *p* < 0.05 was regarded as statistically significant.

## Data Availability

Detailed data will be available upon request from the corresponding author.

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
