# Peer review of "Trends in Long COVID Symptoms in Japanese Teenage Patients"

_medicina, 2023, doi:10.3390/medicina59020261_

Round 1

Reviewer 1 Report

I read with great interest of Sakurada and colleagues. This is an important study as very few tackled long COVID symptomatology in pediatric age group. This makes the study novel of its own and worthy of publication following minor revision.

1. Please include a detailed schematic diagram in the methods section from data gathering to data analysis.

2. Have you characterized patients with long COVID to have received either primary vaccine or booster shot? This is important as patients who received vaccines are at lower risk of experiencing severe long COVID.

3. Several studies have already been published highlighting the different factors contributory to long COVID. Kindy cite these articles:

DOI: 10.1016/j.eclinm.2022.101624

DOI: 10.3390/jcm11247314

DOI: 10.3390/v14122629

Author Response

Responses to Reviewer 1

Comment:

I read with great interest of Sakurada and colleagues. This is an important study as very few tackled long COVID symptomatology in pediatric age group. This makes the study novel of its own and worthy of publication following minor revision.

… Answer: We are very grateful for your evaluation of and interest in our manuscript.  We revised the manuscript point-by-point as you advised below.

Comment:

Please include a detailed schematic diagram in the methods section from data gathering to data analysis.

… Answer: Thank you for your feedback. We added data gathering and analysis flow as Figure 1.

Comment:

Have you characterized patients with long COVID to have received either primary vaccine or booster shot? This is important as patients who received vaccines are at lower risk of experiencing severe long COVID.

… Answer: Thank you for your important suggestion. We additionally compared each group by the number of vaccinations. There was no significant difference between the groups. We added the results and discussion.

Comment:

Several studies have already been published highlighting the different factors contributory to long COVID. Kindy cite these articles:
DOI: 10.1016/j.eclinm.2022.101624
DOI: 10.3390/jcm11247314
DOI: 10.3390/v14122629

… Answer: Thank you for your valuable comments. We cited all of these three articles as you recommended.

Thank you for your constructive review.
We hope that the revised manuscript is now acceptable for publication in Medicina.

Reviewer 2 Report

It is an excellent manuscript, it is understood that, being by the same authors, the wording is very similar to that presented in the article entitled: Application of Kampo Medicines for Treatment of General Fatigue Due to Long COVID, it is recommended to be careful with those sections that are exactly the same in both writings.

It is very interesting to know the clinical manifestations between age groups. Although it was to be expected that the older the age, the greater the complications or the severity of the symptoms, it is necessary that these data be presented and analyzed correctly as in this study. I suggest that the normality in the distribution of the data be presented in order to know if the statistical test used is the most accurate for the analysis.

Author Response

Responses to Reviewer 2

Comment:

It is an excellent manuscript, it is understood that, being by the same authors, the wording is very similar to that presented in the article entitled: Application of Kampo Medicines for Treatment of General Fatigue Due to Long COVID, it is recommended to be careful with those sections that are exactly the same in both writings.

… Answer: We are very grateful for your evaluation of and interest in our manuscript.  Thank you for your important suggestion. We revised the manuscript so as to avoid the same phrasing.

Comment:

It is very interesting to know the clinical manifestations between age groups. Although it was to be expected that the older the age, the greater the complications or the severity of the symptoms, it is necessary that these data be presented and analyzed correctly as in this study. I suggest that the normality in the distribution of the data be presented in order to know if the statistical test used is the most accurate for the analysis.

… Answer: Thank you for your positive feedback. We added detailed description of the statistical analysis steps and normality tests.

Thank you for your constructive review.
We hope that the revised manuscript is now acceptable for publication in Medicina.